# Postmortem Changes in mRNA Expression and Tissue Morphology in Brain and Femoral Muscle Tissues of Rat

**DOI:** 10.3390/ijms26157059

**Published:** 2025-07-22

**Authors:** Sujin Choi, Minju Jung, Mingyoung Jeong, Sohyeong Kim, Dong Geon Lee, Kwangmin Park, Xianglan Xuan, Heechul Park, Dong Hyeok Kim, Jungho Kim, Min Ho Lee, Yoonjung Cho, Sunghyun Kim

**Affiliations:** 1Department of Forensic Science, Graduate School, Catholic University of Pusan, Busan 46252, Republic of Korea; tnwls7859-@naver.com (S.C.); 5585mk@naver.com (M.J.); jutosa70@cup.ac.kr (J.K.); 2Department of Clinical Laboratory Science, College of Health Sciences, Catholic University of Pusan, Busan 46252, Republic of Korea; jungm9911@naver.com (M.J.); hyeongso29@naver.com (S.K.); ehdrjs6789@naver.com (D.G.L.); pkmchi777@naver.com (K.P.); xuan929@naver.com (X.X.); 3Next-Generation Industrial Field-Based Specialist Program for Molecular Diagnostics, Brain Busan 21 Plus Program, Graduate School, Catholic University of Pusan, Busan 46252, Republic of Korea; 4Department of Clinical Laboratory Science, Hyejeon University, Hongseong 32244, Republic of Korea; phc2626@hj.ac.kr; 5Department of Biomedical Laboratory Science, Masan University, Changwon 51217, Republic of Korea; ehd6091@naver.com; 6Forensic DNA Division, Gwangju Institute, National Forensic Service, Jangseong 57248, Republic of Korea; lmh77777@naver.com

**Keywords:** postmortem interval (PMI), histological changes, brain tissues, skeletal muscle tissues, housekeeping gene mRNA

## Abstract

The postmortem interval (PMI), defined as the time elapsed between death and the discovery or examination of the body, is a crucial parameter in forensic science for estimating the time of death. There are many ways to measure the PMI, such as Henssge’s nomogram, which uses rectal temperature measurement; livor mortis; rigor mortis; and forensic entomology. However, these methods are usually affected by various conditions in the surrounding environment. The purpose of the present study was to compare molecular genetics and histological changes in the brain and skeletal muscle tissues of SD rats over increasing periods of time after death. For the PMIs, we considered 0 h, 6 h, 12 h, 24 h, 36 h, 48 h, 4 days, 6 days, 8 days, 10 days, 14 days, and 21 days and compared them at 4 °C and 26 °C. Hematoxylin and Eosin (H&E) staining was performed to observe tissue changes. Morphological tissue changes were observed in cells for up to 21 days at 4 °C, and cell destruction was visually confirmed after 14 days at 26 °C. Total RNA (tRNA) was isolated from each tissue sample, and complementary DNA (cDNA) was synthesized. A reverse transcription quantitative PCR (RT-qPCR) SYBR Green assay targeting three types of housekeeping genes, including *Gapdh*, *Sort1*, *B2m*, and 5S rRNA, was performed. The results showed that *Gapdh* and 5S rRNA were highly stable and could be better RNA targets for estimating the PMI in brain and skeletal muscle tissues. Conversely, *Sort1* and *B2m* showed poor stability and low expression levels. In conclusion, these molecular biomarkers could be used as auxiliary indicators of the PMI in human, depending on the stability of the marker.

## 1. Introduction

Death criteria, including medical evidence, such as personal identification, the cause of death, the time of death, and the type of death, are required to help legal judgment, along with death identification [1].

The postmortem interval (PMI) describes the period of time elapsed from the time of death [2]. To estimate the PMI, the body is checked for changes. Body changes are categorized into early and late body changes. Early body changes typically include a drop in body temperature, hypostasis, and rigor mortis. Late body changes typically include autolysis, putrefaction, the formation of adipocere, mummification, and skeletonization. Today, scientific investigations use a variety of methods to estimate the PMI. These include forensic entomology, the measurement of vitreous humor chemistry, microbiome analysis, metabolomics, proteomics, and, more recently, transcriptomics and RNA degradation analysis [3,4]. Traditional physiological markers, while still in use, are limited by factors such as temperature, humidity, the cause of death, and individual variability, making them less reliable for precise PMI estimation. Molecular approaches—particularly those focused on gene expression profiles and biomolecular stability—have gained increasing attention as complementary or alternative methods that can improve accuracy [3]. Among these, genomic and transcriptomic approaches, such as RNA integrity analysis, microRNA (miRNA) profiling, and mRNA decay, have shown a lot of potential in forensic science due to their sensitivity and temporal correlation with the PMI [5].

The reverse transcription quantitative polymerase chain reaction (RT-qPCR), in particular, enables the quantitative detection of gene expressions and has been widely applied to monitor postmortem mRNA degradation. However, there is a limit to estimating the exact time of death because the surrounding environment, including disease, lifestyle, the temperature at the time of death, humidity, season, and weather, acts as a variable. Brain tissue undergoes changes in which the cells that make up the tissue are destroyed during the PMI [6]. Similarly, skeletal muscle undergoes cell shape distortion and tissue destruction as the PMI progresses. Nevertheless, the rate and pattern of tissue degradation differ by organ. Brain tissue, due to its high lipid content and autolytic enzyme activity, tends to degrade rapidly, whereas skeletal muscle has been shown to preserve structural and molecular integrity longer under the same conditions [7]. This difference influences the reliability of molecular analyses, as RNA from skeletal muscle often remains more stable and extractable at later PMIs compared to that from brain tissue. Thus, muscle tissue may serve as a more suitable candidate for RNA-based PMI estimation, while brain tissue provides unique gene expression information, especially related to neuroimmune markers. The changes in the histology of the human lingual striated muscle make it possible to estimate the PMI, either in the early phase (0–72 h) or the late phase (92–120 h) [8]. The estimation of the PMI by studying RNA decay might be within reach, as RNA degradation or the loss of the RNA transcripts after death seems to be rapid and time-dependent. Housekeeping genes are used in forensic science to estimate the PMI. A study found that RNA is progressively degraded with the PMI and that measuring gene expression in brain tissue at longer PMIs can provide artificially low values [2,9].

Housekeeping genes are commonly described as being essential for cellular existence, regardless of their specific function in the tissue or organism, and are stably expressed irrespective of the tissue type, developmental stage, cell cycle state, or external signal [10]. The *Gapdh* gene is an important glycolytic enzyme that catalyzes the oxidative phosphorylation of glyceraldehyde-3-phosphate to 1,3-diphosphoglycerate [11]. The *B2m* gene encodes the invariant chain of the major histocompatibility complex (MHC) class I. MHC class I molecules, including *B2m*, are now known to be expressed in neurons and to be involved in synaptogenesis and synaptic plasticity [12]. The *Sort1* gene, type I neurotensin receptor-3, is the only known neurotensin receptor expressed on microglial cell lines in the brain, and microglia are responsible for the immunity of brain tissue [13]. The ribosome is a macromolecular assembly responsible for protein biosynthesis in all organisms, and 5S rRNA is a conserved component of the large ribosomal small subunit that is thought to enhance protein synthesis by stabilizing the ribosome structure [14]. A comparison of various tissues in many studies revealed that the degree of gene stabilization is different depending on the tissue type; in particular, *Gapdh* and 5S rRNA have been found to be very stable [15,16].

The RT-PCR refers to a technology that uses a polymerase called polymerase to amplify a specific part or a desired part of synthesized complementary DNA (cDNA) from RNA in large quantities, and qPCR is a technique used to monitor the PCR with a fluorescent signal in real time. In recent forensic DNA investigations, various types of PCR-based analysis tools have been commonly used to identify DNA in very small amounts of trace evidence, and RNA degradation after death is especially useful for precise estimation [17]. The RT-qPCR is considered a good method of choice for detecting mRNA profiles because of its high sensitivity, broad range of equipment usage, and commercially available assays for analysis [18].

In the present study, the PMI consisted of a total of 12 time points, and Spraque-Dawley (SD) rat brain and skeletal muscle tissues were collected under two temperature conditions: the average winter temperature (4 °C) and the average summer temperature (26 °C). Histological changes were observed in the tissues using hematoxylin and eosin (H&E) staining under light microscopy. Total RNA (tRNA) was isolated and quantified, and three types of housekeeping genes, including *Gapdh*, *Sort1*, *B2m*, and 5S rRNA, were analyzed by the RT-qPCR SYBR Green assay, according to the PMI.

## 2. Results

### 2.1. The Change in Tissue Morphology in the Brain Cortex, Hippocampus, and Cerebellum, According to the PMI and Under Different Temperatures

The cerebral cortex, hippocampus, and cerebellum of SD rats was observed under an optical microscope (Figure 1). Then, the cell changes between before and after death were magnified and compared.

The cerebral cortex area has pyramidal cells. Pyramidal cells, as their name suggests, are triangle-shaped cells that integrate sensory nerve information and initiate voluntary motor responses. Regarding the changes in brain tissues according to the PMI, pyramidal neurons (P), axons (A), and oligodendroglia (O) were assessed (Appendix A). The morphology of these structures showed a clear difference depending on the temperature. There were no cell changes at 4 °C from day 0 to day 21. By contrast, at 26 °C, cytoplasmic and cell destruction were visually confirmed from day 14 (Appendix A).

The morphological results of the hippocampal area in brain tissues are shown in Figure 1B. The hippocampus was divided into cornu ammonis 1 (CA1), cornu ammonis 2 (CA2), and cornu ammonis 3 (CA3) sections (Figure 1B), and the cells in these sections were observed (Figure 1B, Appendix A). Similar to the brain cortex, clear differences appeared in the hippocampus, depending on the temperature over time. At 4 °C, the number of cells decreased over time; however, there were no changes in the cell shape. On the other hand, at 26 °C, the deformation of the cytoplasm and nucleus was visually confirmed from day 4. On day 14, cell nuclei were not found, and only the shape of the cells was observed. On day 21, all the cells collapsed, and only the cytoplasm could be observed.

The morphological results of the cerebellum in brain tissues are shown in Figure 1C. The cerebellum area of the brain includes Purkinje neurons (P), the medulla of white matter (M), a granular layer (GL), and a molecular layer (ML) (Appendix A). At 4 °C, the shape of the Purkinje cells and the distinction between the molecular layer and the granular layer were clear until day 21. At 26 °C, each layer could be distinguished until the 4th day, but from day 14, the cells in each layer had been destroyed and the boundaries of the cell layers were unclear. On day 21, the cells became distorted, and the shape of the cells could not be deciphered.

### 2.2. Change in Tissue Morphology in Longitudinal Sections and Transverse Sections of Skeletal Muscle Tissues According to PMI Under Different Temperature Conditions

Morphological changes in the skeletal muscle tissue were observed in longitudinal and transverse sections (Figure 2), showing a decrease in cells and changes in cell morphology over time. The skeletal muscle was examined for striations, nuclei (N), connective tissue, and muscle fibers (Appendix A). Regarding the longitudinal section of muscle tissue, the majority of cells were intact from day 0 to day 21 at 4 °C. At 26 °C, a decrease in the number of cells and cell changes was observed from day 4, and only the shape of the muscle fibers was observed from day 14 to day 21 (Figure 2A). In the transverse section, you can see that the cells and connective tissue distributed around the muscle fibers were visible. In this region, at 4 °C, the cells around the muscle fibers decreased over time, but the cell shape was maintained (Figure 2B). At 26 °C, the size of the cells decreased, and their shape began to change from day 4. Only the shape of the muscle fibers was confirmed from day 14 (Figure 2B).

### 2.3. The Ct Values of Genes in the Brain Tissues and Skeletal Tissues Identified Using an RT-qPCR SYBR Green Assay, According to the PMI Under Different Temperature Conditions

These results showed that the cycle threshold (Ct) values of three target genes and one reference gene, including *Gapdh*, *B2m*, *Sort1*, and 5S rRNA, in brain tissues clearly decreased depending on the gene, according to the RT-qPCR SYBR Green analysis. For more accurate results, after performing the RT-qPCR experiments, all the threshold values were set to be the same. All the threshold values of the target genes and reference gene were set to 354 relative fluorescence units (RFUs).

Further, 5S rRNA was amplified earlier in the brain tissues, followed by *Gapdh*, *Sort1*, and *B2m*. The 5S rRNA was amplified after around 10 cycles in the brain tissues, except in the day 21 and 4 °C samples. *Gapdh* was amplified after around 20 cycles in the brain tissues. *Sort1* was amplified in 20 cycles for the 0 h to 48 h samples; however, the samples after day 4 were amplified by around 30 cycles in the brain tissues. *B2m* was amplified over 30 cycles in brain tissues. Based on the results of the RT-qPCR SYBR Green assay in brain tissues, the Ct values of 5S rRNA were indicated to be the most reliable for estimating the PMI (Figure 3, Appendix A).

The results of the RT-qPCR SYBR Green assay targeting three target genes and one reference gene in skeletal muscle tissue showed the same pattern as the results in brain tissue. The 5S rRNA was amplified earlier in the brain tissues, *B2m* was amplified later, and temperature did not act as a factor affecting the experimental results. Similarly, the Ct values of 5S rRNA in the brain tissues were amplified in around 10 cycles, except in the day 14 and 26 °C samples. *Gapdh* was amplified by around 20 cycles in the skeletal muscle tissues. *Sort1* was amplified in around 20 cycles for the 0 h and 48 h samples; however, the samples after day 4 were amplified by around 30 cycles in skeletal muscle tissues. *B2m* was amplified after over 30 cycles in skeletal muscle tissues. Based on the results of the RT-qPCR SYBR Green assay in skeletal muscle tissues, the Ct values of 5S rRNA were indicated to be the most reliable for estimating the PMI (Figure 4, Appendix A).

In RT-qPCR, the Ct value represents the number of amplification cycles required for the fluorescence signal to cross a defined threshold. Therefore, a lower Ct value indicates a higher initial abundance of RNA, and conversely, a higher Ct value indicates a lower RNA quantity due to degradation.

Although the original text describes that “Ct values decreased,” this interpretation should be clarified: it does not necessarily imply stability over time but rather reflects the relative abundance of transcripts at each time point. Thus, increased Ct values at later PMI points are expected due to RNA degradation. In the present study, Ct values were interpreted in this context.

### 2.4. The Delta Ct Values of Target Genes in the Brain Tissues and Skeletal Tissues Identified by RT-qPCR SYBR Green Assay, According to the PMI and Under Different Temperature Conditions

To account for differences in the initial transcript abundance and to better reflect mRNA degradation over time, ∆Ct values were calculated using 5S rRNA as the reference gene for each target gene (*Gapdh*, *Sort1*, and *B2m*). A higher ∆Ct value indicates reduced target gene expression relative to the stable 5S rRNA and, therefore, may reflect increased degradation over time.

Figure 5A (Appendix A) shows the ∆Ct values of the three target genes in brain tissues. Sort1 exhibited a clear increase in ∆Ct over time at 26 °C, particularly after day 6, suggesting substantial degradation relative to the reference gene. In contrast, *Gapdh* and *B2m* maintained relatively stable ∆Ct values across all the time points and temperatures, implying greater postmortem stability.

Figure 5B (Appendix A) presents the ∆Ct values in skeletal muscle tissue. All three genes (*Gapdh*, *Sort1*, and *B2m*) showed relatively constant ∆Ct values, regardless of the postmortem time or temperature. This suggests that these genes are more stable in muscle tissue, or that the degree of degradation was not sufficient to significantly affect their relative expression over the tested PMIs.

These ∆Ct data indicate that while 5S rRNA appears to be a reliable reference gene, only *Sort1* in brain tissue showed a consistent and interpretable increase in ∆Ct with the increasing PMI at the higher temperature, which could be used as a potential molecular marker for PMI estimation.

## 3. Discussion

Research on estimating the PMI has remained active. The postmortem degradation of nucleic acids has been suggested as an elegant alternative to classical methods for PMI estimation [9]. The PMI refers to the passage of time after death. As the PMI is a very sensitive topic, much effort is being put into forensic science today to estimate the PMI more accurately.

Housekeeping genes are used to estimate the PMI in forensics. Studies have shown that RNA is progressively degraded as the PMI increases, and that measuring gene expression in brain tissue as the PMI lengthens can provide artificially low values [5]. Housekeeping genes are essential for cell existence and are always stably expressed [10]. The *Gapdh* gene is an important glycolytic enzyme that catalyzes the oxidative phosphorylation of glyceraldehyde-3-phosphate to 1,3-diphosphoglycerate [9]. In addition, MHC class I molecules, including *B2m*, are now known to be expressed in neurons and involved in synaptogenesis and synaptic plasticity [12]. *Sort 1*, a type I neurotensin receptor-3, is responsible for the immunity of brain tissue [13]. The 5S rRNA molecule is a conserved component of the large ribosomal small subunit that plays a role in stabilizing the ribosome structure and, thus, in enhancing protein synthesis, and it is always present within the ribosome [14].

To compare postmortem mRNA degradation, we examined changes in gene expression distribution in the short-term PMI (S-PMI) and the long-term PMI (L-PMI) groups across all the tissues and found that most of the genes were decreased in the L-PMI. This experiment was conducted based on previous research showing that mRNA degradation increases as the PMI increases [19]. Based on various previous studies, the present study compared and analyzed the expression of mRNA over time after death using the four housekeeping genes, 5S rRNA, *Gapdh*, *Sort1*, and B*2m,* using the SD rat brain and the right and left femoral muscles as samples.

To estimate the postmortem elapsed time, the temperature was divided into two groups, 4 °C and 26 °C, and the progress was observed by storing the samples in an airtight container to block other environmental variables. The mRNA expression was analyzed in the brain and femoral muscles under the same conditions using housekeeping genes, whose stability was analyzed in several experiments. The results showed clear differences in housekeeping gene expression were confirmed in the results of the present study. The RT-qPCR SYBR Green assay targeting housekeeping gene expression was performed. Comparing Ct values, 5S rRNA showed the lowest value in both brain and thigh muscle tissue, followed by *Gapdh*, *Sort1*, and B*2m.* Comparing the Ct values, we can see that *Sort 1* is a gene related to the brain’s immune system [13]; thus, its Ct value was relatively low in brain tissue compared to femoral muscle tissue. Furthermore, *B2m* showed the highest Ct value, and there was no effect of temperature on gene expression, depending on the time elapsed after death. The stability of *Gapdh* and rRNA has already been proven in several studies. *Gapdh* is a stable gene in the brain and skeletal muscle; thus, such a result was expected to be derived [19,20].

When compared by temperature, the results of this study showed that temperature did not affect the PMI, but another study showed that in the group where the temperature was adjusted to 4 °C, ΔCt values for both target genes increased within 48 h after euthanasia. It was found to show generalized stability with the PMI indicated. Further, the ΔCt value of the 26 °C group was stable for the first 2 h; however, it decomposed from 2 h to 24 h and appeared to be correlated with the PMI [21]. Additionally, our finding that intact RNA was abundant in tissues frozen for more than 36 h after death is consistent with the results of a previous study [22]. The mRNA degradation could be influenced by external factors, such as sunlight, humidity, or a high temperature during the PMI, as well as internal parameters, the such as tissue type, cause of death, sex, age at death, and specific pre-death conditions [23]. Based on these results and comparing them with the results of this study, it is believed that there would have been meaningful results if the temperature conditions had been set more broadly.

To estimate the PMI, we compared and analyzed mRNA to identify discover stable RNA biomarkers in brain and femoral muscle tissues. Through a comparison of the postmortem period and Ct values for *Gapdh* mRNA in the rat brain using the real-time PCR method, it was confirmed that the Ct value increased with time [24]. Our analysis of the muscle tissue Ct value showed a decrease over time. In a previous study that investigated the normalization of reference genes in brain tissue, *Gapdh* and *Sort1* showed relatively high stability [16], 5S rRNA and *Gapdh* were the most reliable gene candidates for reference genes in heart and brain tissue, and *B2m* showed unreliable results [13]. Based on these results, a comparison of the results of this experiment showed that the four housekeeping genes were also similar to those of previous studies; thus, the results were considered a meaningful result.

However, it should be noted that Ct values primarily reflect the abundance of the starting template, rather than the degradation status alone. In our study, 5S rRNA consistently showed the lowest Ct values, suggesting high baseline expression, rather than necessarily slower degradation. To assess the RNA stability more appropriately, we calculated the ∆Ct values using 5S rRNA as an internal control. In brain tissue, Sort1 showed an increasing ∆Ct over time at 26 °C, indicating relative degradation, whereas *Gapdh* and *B2m* remained relatively stable. This implies that Sort1 may be more suitable as a PMI-sensitive marker under elevated temperatures. However, these trends were not uniformly observed in skeletal muscle. Although we present overall gene expression trends, the interpretation of ∆Ct values in terms of the PMI could benefit from the inclusion of representative numerical values or graphs, which we plan to include in future studies for better clarity.

The entire brain was sampled, and the femoral muscles were sampled from the area closest to the femur. Microscopic results obtained by H&E staining were affected by temperature. In both brain and thigh muscle tissue, there was little cell change until day 21 at 4 °C, but cell reduction, changes in cell shape, and cell destruction were clearly observed from day 4 at 26 °C (Figure 2). These results are the same as those of previous studies showing that cells change over time [6,8].

A limitation of the present study is that the integrity of isolated tRNA was not analyzed. cDNA was synthesized by adjusting the RNA concentration to 1 µg, and the experiment was conducted using the same Ct value as RT-qPCR. More meaningful experimental results could be obtained if tRNA integrity was measured using a bioanalyzer as a follow-up study.

In addition to this, the integrity of the total RNA was not assessed using the RNA integrity number (RIN), which is widely used to evaluate RNA quality. Although the RNA quantity was standardized prior to cDNA synthesis, the absence of RIN evaluation may limit the interpretability of the gene expression data, particularly in the context of postmortem degradation. RIN is known to correlate with mRNA stability and has been recommended in various studies to enhance the reliability of gene expression analysis in forensic investigations [25]. Future studies should incorporate RIN measurements using an RNA bioanalyzer to better assess the RNA quality and support the validity of molecular findings.

Furthermore, although four housekeeping genes were used, including 5S rRNA, their degradation patterns may differ substantially from that of mRNA due to structural differences. For example, 5S rRNA is highly stable and may not accurately reflect mRNA degradation kinetics over time. This mismatch can introduce a bias during normalization and potentially distort the estimation of the PMI. It is also important to note that using a single reference gene without validating its stability under all the experimental conditions could further contribute to variability in the results. As recommended in the literature [26], normalization strategies should involve multiple validated reference genes with similar degradation behavior to the target genes. These considerations highlight important limitations and should be addressed in future studies to strengthen the forensic applicability of mRNA-based PMI estimation.

Although this study demonstrated that mRNA degradation patterns in rat brain and femoral muscles can be used to estimate the PMI, caution should be exercised when generalizing these findings to human samples. The present experiment was conducted using 6-week-old, genetically homogeneous SD rats under controlled conditions with uniform causes of death and no age variation. However, human forensic cases are more complex and influenced by diverse factors, such as the genetic background, age at death, cause of death, and environmental variables (e.g., temperature, humidity, microbial exposure).

Moreover, previous studies indicate that RNA degradation in human tissues does not always correlate linearly with the PMI, and that brain and muscle are among the most RNA-preserving tissues postmortem [27]. These interspecies differences mean that our conclusions, while promising in a controlled animal model, require careful validation in human forensic cases before being applied clinically. Further studies using human tissue under variable environmental and biological conditions are necessary to confirm the forensic utility of the identified RNA biomarkers.

In conclusion, our findings demonstrate the potential utility of mRNA degradation profiles—particularly in brain tissue—as molecular indicators for estimating the PMI. The observed gene- and tissue-specific expression patterns, along with histological validation, provide a foundation for integrating molecular and structural markers in forensic investigations. Future research should focus on validating these findings in human tissues across diverse conditions, including different ages, causes of death, and environmental exposures. Incorporating advanced RNA quality metrics, such as RIN, as well as exploring new reference genes with human-specific stability, will be essential to translate these results into robust forensic tools applicable in real-world scenarios.

## 4. Materials and Methods

The animal protocol used in the present study was reviewed by the Catholic University of Pusan-Institutional Animal Care and Use Committee (CUP-IACUC) on their ethical procedures and scientific care, and it was approved (Approval number CUP AEC 2023-001). A total of 72 6-week-old male SD rats (Kosa Bio Inc., Seongnam, Republic of Korea) used in this study were randomly divided into 24 groups (three rats per group). Each of the three animals in a group was processed and analyzed individually, and no pooling of biological samples was performed at any stage of the experiment. In accordance with the ethics of animal experimentation and previous studies showing that protease inhibitor administration immediately before death has a significant effect on postmortem muscle deformation [28], the animals were asphyxiated using a CO_2_ chamber to exclude interference. Temperature conditions were divided into Korea’s average summer temperature, 26 °C, and the winter average temperature, 4 °C. Tissue samples were collected at 0 h, 6 h, 12 h, 24 h, 36 h, 48 h, 4 days, 6 days, 8 days, 10 days, 14 days, and 21 days.

The organ tissue samples were immersed in a 10% formalin solution for 24 h for the fixation and preservation of cellular and structural components. The fixed organ sample was cut to the desired size and processed to pass through paraffin wax. Paraffin wax is used to harden organ samples and as the main material for making paraffin blocks. Organ sections treated with paraffin wax were used to make blocks for organ cutting and thin organ sections were made from paraffin blocks. Sections of these organs were mounted on slides and stained. The organ slides were then exposed to H&E stain. Hematoxylin stains the cell nucleus blue, and eosin stains the cell structure and cytoplasm pink. This staining process allows for the visual differentiation of the cellular and structural elements of the organ. The stained organ slides were then observed using a Leica S9D stereo microscope (Leica Microsystems, Wetzlar, Germany) at 40× and 400× magnification.

The samples included brain and femoral muscles collected from the rats. Half of the brain was used to identify histological changes, and the other half was subjected to tRNA extraction. The hippocampus, cortex, and cerebellum regions of the brain were included to identify changes in the tissues.

The femoral muscle was collected from the area closest to the femur based on experimental results showing that the higher the lipid content, the more difficult it is to obtain a good quality sample [29]. Half of the muscle samples were used to identify tissue changes, and the remaining half was subjected to RNA extraction. The tRNA isolation from stabilized rat organ tissue samples was performed using the RiboEx^TM^ (GeneAll, Seoul, Republic of Korea). The tissue sample was weighed and cut into 0.01 g pieces of the same size. The sample was homogenized with 1 mL of RiboEx^TM^ lysis solution (GeneAll). Next, 0.2 mL of chloroform was added, and the sample was incubated for 2 min at room temperature. The sample was centrifuged at 12,000× *g* for 15 min at 4 °C. After centrifugation, 0.5 mL of isopropyl alcohol was added, and the samples were incubated for 10 min at room temperature. The sample was centrifuged at 12,000× *g* for 10 min at 4 °C, and the supernatant was discarded. To wash the RNA pellet, 1 mL of 75% ethanol was added and centrifuged at 7500× *g* for 5 min. The supernatant and the ethanol were carefully discarded, and the RNA pellet was air-dried for 5 min. The RNA was dissolved in DEPC-treated water by incubating for 10 min at 56 °C. The concentration and purity of the isolated tRNA were measured using a NanoDrop 2000 spectrophotometer (Thermo Fisher Scientific, Waltham, MA, USA), and the samples were stored at −80 °C until further use.

cDNA was synthesized using a Moloney Murine Leukemia Virus (MMLV) Reverse Transcriptase kit (Invitrogen, Carlsbad, CA, USA) according to the manufacturer’s instructions. Before the procedure, the total RNA was diluted to 100 ng/uL. First, 10 uL of tRNA was added to a master mix containing 1 uL of 10 mM dNTP mix (10 mM each of dATP, dGTP, dCTP, and dTTP at a neutral pH), 1 uL of DEPC-treated water, and 1 uL of a random hexamer (Invitrogen) in a PCR tube. Second, the reaction mixture was incubated at 65 °C for 5 min before being quickly chilled on ice and spun down. Third, a mixture of 4 uL of 5× First-Strand Synthesis Buffer, 2 uL of 100 mM dithiothreitol (DTT), and 1 uL (1 unit) of M-MLV RT were added to the previous reaction mixture in PCR tubes and then incubated at 25 °C for 10 min, 37 °C for 50 min, and 70 °C for 15 min. All the reactions were performed using a SimpliAmp thermal cycler (Life Technologies, Carlsbad, CA, USA). The synthesized final cDNA samples were stored at −20 °C until further use.

Each reaction in the PCR plates contained 10 uL of SYBR Green Real-Time PCR Master Mix (TOYOBO, Osaka, Japan), 1 uL each of 10 pmol sense and anti-sense oligonucleotide primer sets, 5 uL of ultra-pure DNase/RNase-free distilled water, and 3 uL of a synthesized cDNA template in a final volume of 20 uL. In the case of RT-qPCR, the positive control group was tested with a sample of 0 h, and the negative control group was tested using DNase/RNase-free distilled water. The RT-qPCR primer pairs were used to detect 5S rRNA and three types of housekeeping gene mRNAs: *B2m*, *Gapdh*, and *Sort1* (Table 1). The thermal cycling conditions were 60 s at 95 °C, followed by 35 cycles of 10 s at 95 °C and 30 s at 60 °C. For the relative expression of each target marker, we examined the quality of the RNA by determining the cycle threshold (Ct), meaning the number of PCR cycles required for the fluorescence to be significantly higher than the background. The fluorescence detection threshold was uniformly set at 354 RFU for all the genes to maintain consistency and avoid background signal interference. This threshold was selected based on preliminary tests that ensured reliable amplification while minimizing noise, in accordance with the manufacturer’s recommendations.

The primer efficiency was assessed by generating standard curves with serial dilutions of cDNA, and all the primers showed acceptable efficiency (90–110%). The specificity of the amplification was confirmed by a melting curve analysis, showing a single sharp peak for each primer set, confirming the absence of non-specific amplification.

The ΔCt values were calculated using 5S rRNA as the reference gene, according to the following formula: ΔCt = Ct (target gene) − Ct (5S rRNA).

The statistical analysis was performed using GraphPad Prism v. 5.00 (GraphPad Software, San Diego, CA, USA). Statistically significant differences in outcomes were calculated to compare the tissue, temperature, PMI, and housekeeping genes. The Ct value, which is the result of charging the RT-qPCR, was compared using one group relative to the other two groups. The ∆Ct value was used to compare the remaining three primers (*B2m*, *Gapdh*, and *Sort1*) relative to 5S rRNA. The ∆Ct values were compared using a group used to replicate values in side-by-side sub columns relative to the PMI and also using a column used to replicate values, stacked into columns. All the data are expressed as the mean ± standard deviation (SD), and statistical significance was determined using a one-way ANOVA followed by a Tukey’s multiple comparison test. A *p*-value of less than 0.05 (*p* < 0.05) was considered statistically significant.

## Figures and Tables

**Figure 1 ijms-26-07059-f001:**
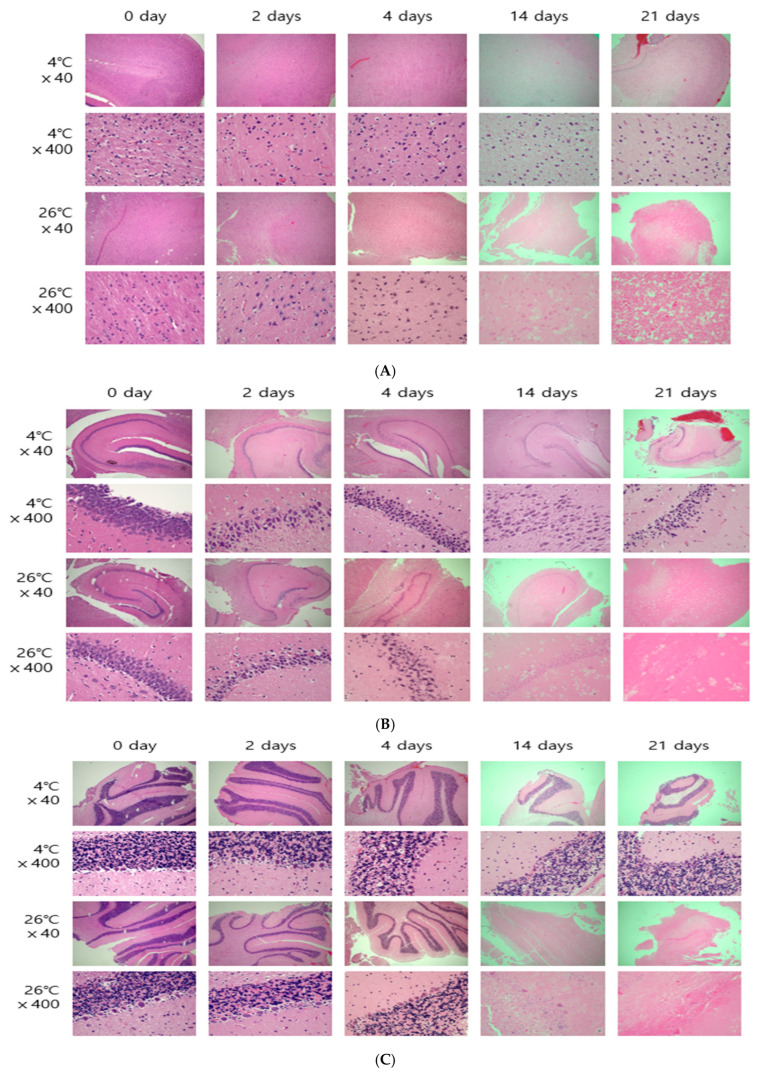
Histological changes in the brain cortex (**A**), hippocampus (**B**), and cerebellum (**C**) according to the PMI, observed using light microscopy after H&E staining (×40, ×400).

**Figure 2 ijms-26-07059-f002:**
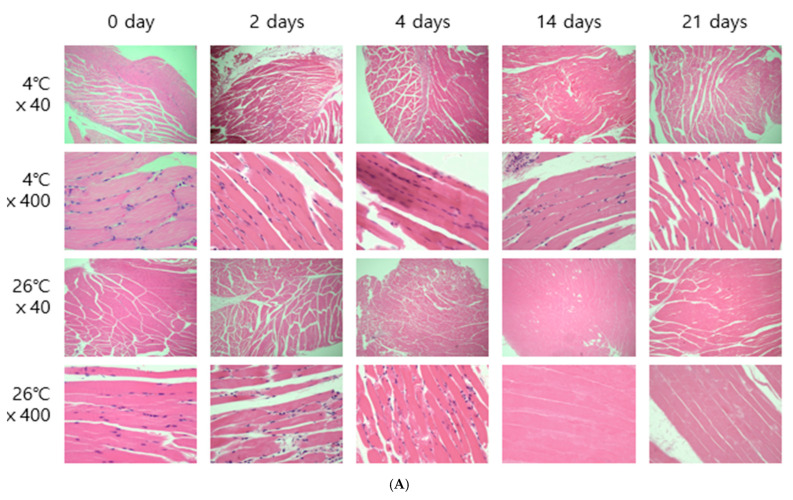
Histological changes in longitudinal sections (**A**) and transverse sections (**B**) of the femoral muscle tissues according to the PMI, observed using light microscopy after H&E staining (×40, ×400).

**Figure 3 ijms-26-07059-f003:**
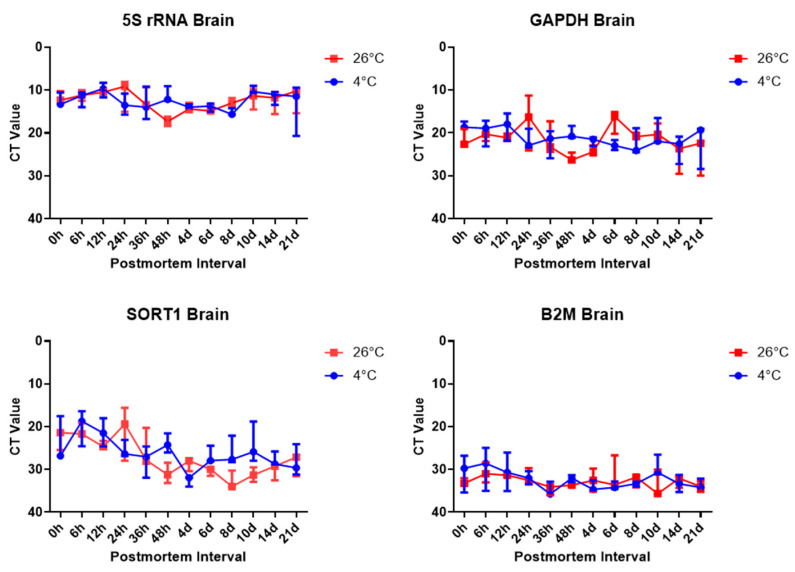
Ct values of target genes and the reference gene, 5S rRNA, *Gapdh*, *Sort1*, and *B2m*, in brain tissues, according to the PMI.

**Figure 4 ijms-26-07059-f004:**
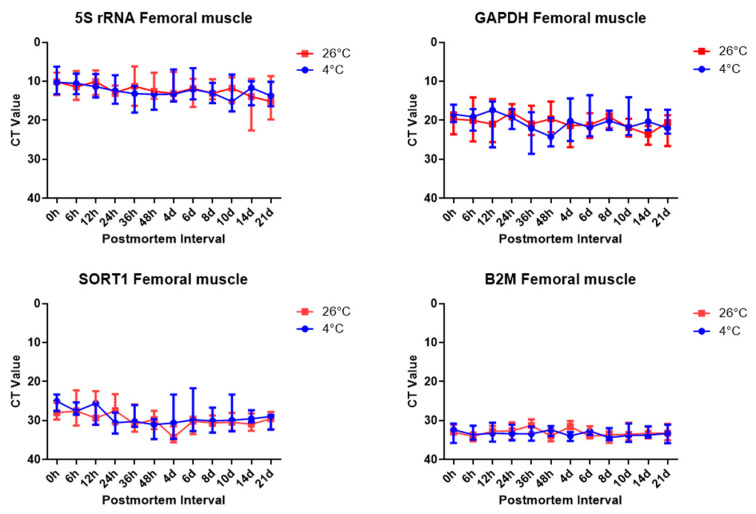
Ct values of target genes and the reference gene, 5S rRNA, *Gapdh*, *Sort1*, and *B2m*, in skeletal muscle tissues, according to the PMI.

**Figure 5 ijms-26-07059-f005:**
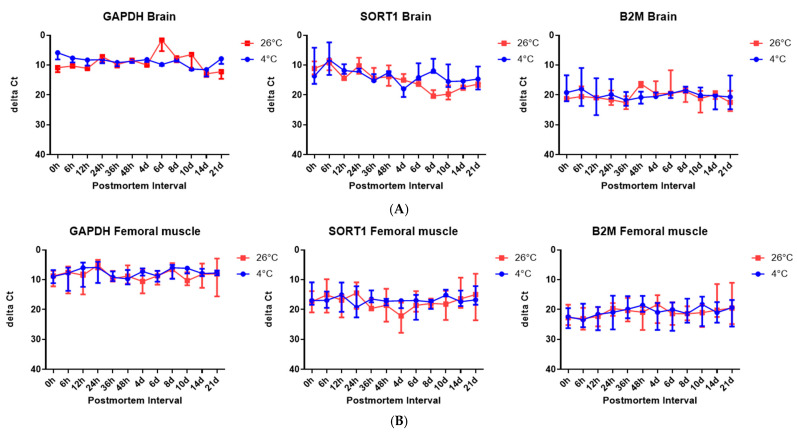
The ∆Ct values of the target genes—*Gapdh*, *Sort1*, and *B2m*—in brain tissues, according to the PMI, (**A**) and in skeletal muscle tissues, according to the PMI (**B**).

**Table 1 ijms-26-07059-t001:** Oligo-nucleotide primer pairs for the RT-qPCR SYBR Green assay used in the present study.

Genes	Gene Description	Forward Primer (5′→3′)	Reverse Primer (5′→3′)	Gene Accession No.	Amplicon Size (bp)	References
5S rRNA	5S Ribosomal RNA	ATCTCGTCTGATCTCGGAA	TCTCCCATCCAAGTACTAACC	k01594	59	[15]
*B2m*	Beta-2 microglobulin	TGACAACTTTGGCATCGTGG	GGGCCATCCACAGTCTTCTG	NM_017008	78	
*Sort1*	Sortilin 1	AGTAGGAGGTGCTCGATGAAG	TCCTGTAGAGCAGCAACAGG	NM_031144	148	
*Gapdh*	Glyceraldehyde-3-phosphate dehydrogenase	CTGACCAACAATACGCACCA	AGTTCTCGGGACCAATAGCC	XM_342317	250	[16]

## Data Availability

The additional data supporting the manuscript are available from the corresponding author upon request.

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
