# Peer review of "Postmortem Changes in mRNA Expression and Tissue Morphology in Brain and Femoral Muscle Tissues of Rat"

_ijms, 2025, doi:10.3390/ijms26157059_

Round 1
Reviewer 1 Report
Comments and Suggestions for Authors
Dear Authors, here are some comments and suggestions that should be incorporated into the manuscript. The topic addressed is of great importance, but an extensive revision of the manuscript is required.
- The definition of PMI is incomplete: “ Post-mortem interval (PMI) is the time that has elapsed since an individual’s death” … and the first observation of the corpse or the corpse discovery.
- “There are many ways to measure PMI such as rectal temperature measurement …” the rectal temperature alone is not sufficient for PMI estimation (Henssge nomogram). Authors have to correct the sentence.
- The introduction is lacking. The authors refer to the most commonly used and recognized methods by the scientific community for PMI estimation, but in an imprecise and too superficial way. It is advisable to better define these aspects to avoid imprecise generalizations. I suggest adding articles that refer to all these aspects (doi: 10.3390/ijms25179207; doi: 10.1007/s12024-016-9776-y).
- References are not in order (1->4->2), so they need to be modified. The same goes for images (figures 1–10), which must be cited in the text in order.
- Rigor mortis is written in italics; it should be modified.
- If a sentence cites more than one reference, all the references should be put into the same brackets (es. Line 63, ref 4 and 5; line 78, etc).
- Definitions should be always the same (e.g. “target gene”, “reference gene”): if 5S is a reference gene, always use the same term throughout the text.
- The postmortem intervals investigated are numerous, and the tissues studied (brain and femoral muscle) were used both for histology and for RNA extraction, a methodologically adequate choice to compare structural and molecular integrity. Despite that, no mention is made of quality controls such as RIN (RNA Integrity Number) on the extracted RNAs, as the authors consider it only in a future perspective. RIN measures would have strengthened the validity of the molecular analysis, as highlighted in the literature. Furthermore, the use of 5S rRNA and a single reference gene may lead to bias, since 5S and mRNA have different degradation dynamics. The authors should therefore add a paragraph "Risk of Bias" or "Limitations", or deepen these aspects in the discussion, in light of what is described in the literature. The authors are suggested to consider the importance of normalization, RIN… (doi: 10.1186/1471-2105-11-253; doi: 10.3390/ijms25158185).
- The authors used 72 6-week-old Sprague-Dawley rats, divided into 24 groups of 3 animals each, however, it is not clear whether the three animals per group were analyzed individually or in pools.
- The authors do not mention the sex difference in the sample (how many males/females?).
- In the “methods” section, authors could indicate how long the samples remained in formalin for fixation.
- Have primer efficiency tests or dissociation curve analyses been performed? Authors should explicit it in the text. The statistical analysis is described in poorly understandable terms and there is a lack of reference to threshold values ​​(e.g. p<0.05). Authors should at least explain why.
- The detection threshold was set at 354 RFU for all genes, but it is not explained how this threshold was selected or whether it is satisfactory. The statistical description (GraphPad Prism) and how ∆Ct was calculated is unclear and fragmented, no mean values ​​± deviations, nor p-values ​​or standard errors are reported. Authors should clarify these aspects.
- The authors state (Results 2.3) that 5S is always amplified first (Ct about 10), followed by Gapdh (~20), Sort1 and B2m (the latter Ct >30). They suggest that low Ct indicates high stability. This reasoning is sensible but not complete: a low Ct indicates more initial template, therefore 5S is more abundant. To estimate the PMI it would be more correct to use ∆Ct (normalizing to a constant reference gene) or ΔΔCt between different times. The authors calculate a ΔCt using 5S as a reference but then do not clarify how to interpret the resulting ∆Ct in terms of PMI. For example, they say that in brain the ∆Ct of Sort1 varies with time at 26°C while Gapdh and B2m, while in muscle “no PMI effect” is observed. These results are confusing: if 5S is stable, but Sort1 is decreasing, one would expect a large increase in the ΔCt of Sort1/5S. However, the discussion does not show any concrete graphs or values, making it difficult to evaluate. Authors should clarify these aspects.
- Authors state that these biomarkers could also be used in human tissues to estimate PMI, based on relative stability, but this generalization is not completely supported. The data are obtained in 6-week-old homozygous rats, with no age variations or causes of death. Knowing that species, tissues and environmental context greatly influence degradation, it is not automatic to translate the results to the human system. For example, studies indicate that in human samples PMI does not always linearly influence RNA quality (doi: 10.1007/s00414-006-0131-9) and that muscle and brain are among the best preserved tissues. The authors should rephrase this concept.
- There are some arbitrary statements. For example, the discussion states that 5S “prevents RNA degradation by intracellular exonucleases”. This statement is vague and biologically imprecise; it is more correct that ribosomal RNA is abundant, not that it “prevents” degradation. Furthermore, a difference in 5S Ct between 26°C and 4°C “in kidney and liver” after 48h is cited, but the study does not show kidney or liver; the authors are probably mixing their own data and literature. Such misleading details should be corrected or clarified (e.g. indicating whether they derive from previous studies or from the experimental set not shown).
- The terms “Ct decreases” and “increases” are confusing in the text. In qPCR a lower Ct indicates more RNA, but in post-mortem data one would expect higher Ct (degradation) with increasing PMI. The paper states that “Ct values ​​of target genes decreased”, which seems to be a conceptual error. Additionally, the statistical method described for comparing Ct/∆Ct is unclear. If authors do not agree with this interpretation, they should tell me why and clarify it in the text.
The English language needs a thorough revision. There are grammatical and sentence formulation errors, which make the text difficult to understand and reduce its fluency.
Author Response
Comments 1: The definition of PMI is incomplete: “ Post-mortem interval (PMI) is the time that has elapsed since an individual’s death” … and the first observation of the corpse or the corpse discovery.
Response 1: Thank you for pointing this out. We agree with this comment. Therefore, we have changed to emphasized this point. Mention exactly where in the revised manuscript this change can be found - 1 page, 1 paragraph, and 24-26 line.
Comments 2: “There are many ways to measure PMI such as rectal temperature measurement …” the rectal temperature alone is not sufficient for PMI estimation (Henssge nomogram). Authors have to correct the sentence.
Response 2: Thank you for pointing this out. We agree with this comment. Therefore, we have changed to emphasized this point. Mention exactly where in the revised manuscript this change can be found - 1 page, 1paragraph, and 27 line.
Comments 3: The introduction is lacking. The authors refer to the most commonly used and recognized methods by the scientific community for PMI estimation, but in an imprecise and too superficial way. It is advisable to better define these aspects to avoid imprecise generalizations. I suggest adding articles that refer to all these aspects (doi: 10.3390/ijms25179207; doi: 10.1007/s12024-016-9776-y).
Response 3: Thank you for pointing this out. We agree with this comment. Therefore, we have changed to emphasized this point. Mention exactly where in the revised manuscript this change can be found - 2 page, 2 paragraph, and 55-65 line.
Comments 4: References are not in order (1->4->2), so they need to be modified. The same goes for images (figures 1–10), which must be cited in the text in order.
Response 4: Thank you for pointing this out. We agree with this comment. Therefore, we have changed to emphasized this point. Mention exactly where in the revised manuscript this change can be found - 2 page, 2 paragraph, and 50-58 line.
Comments 5: Rigor mortis is written in italics; it should be modified.
Response 5: Thank you for pointing this out. We agree with this comment. Therefore, we have changed to emphasized this point. Mention exactly where in the revised manuscript this change can be found - 2 page, 2 paragraph, and 53 line.
Comments 6: If a sentence cites more than one reference, all the references should be put into the same brackets (es. Line 63, ref 4 and 5; line 78, etc).
Response 6: Thank you for pointing this out. We agree with this comment. Therefore, we have changed to emphasized this point. Mention exactly where in the revised manuscript this change can be found - 2 page, 2 paragraph, and 58 line. 3 page, 1 paragraph, and 103 line.
Comments 7: Definitions should be always the same (e.g. “target gene”, “reference gene”): if 5S is a reference gene, always use the same term throughout the text.
Response 7: Thank you for pointing this out. We agree with this comment. Therefore, we have changed to emphasized this point. Mention exactly where in the revised manuscript this change can be found - 4 page, 3 paragraph, and 170-171 line. 4 page, 3 paragraph, and 174 line. 4 page, 5 paragraph, and 184-185 line. 8 page, 1 paragraph, and 238 line. 8 page, 2 paragraph, and 241 line.
Comments 8: The postmortem intervals investigated are numerous, and the tissues studied (brain and femoral muscle) were used both for histology and for RNA extraction, a methodologically adequate choice to compare structural and molecular integrity. Despite that, no mention is made of quality controls such as RIN (RNA Integrity Number) on the extracted RNAs, as the authors consider it only in a future perspective. RIN measures would have strengthened the validity of the molecular analysis, as highlighted in the literature. Furthermore, the use of 5S rRNA and a single reference gene may lead to bias, since 5S and mRNA have different degradation dynamics. The authors should therefore add a paragraph "Risk of Bias" or "Limitations", or deepen these aspects in the discussion, in light of what is described in the literature. The authors are suggested to consider the importance of normalization, RIN… (doi: 10.1186/1471-2105-11-253; doi: 10.3390/ijms25158185).
Response 8: Thank you for pointing this out. We agree with this comment. Therefore, we have changed to emphasized this point. Mention exactly where in the revised manuscript this change can be found - 11 page, 5 paragraph, and 348-352 line.
Comments 9: The authors used 72 6-week-old Sprague-Dawley rats, divided into 24 groups of 3 animals each, however, it is not clear whether the three animals per group were analyzed individually or in pools.
Response 9: Thank you for pointing this out. We agree with this comment. Therefore, we have changed to emphasized this point. Mention exactly where in the revised manuscript this change can be found - 12 page, 5 paragraph, and 400-403 line.
Comments 10: The authors do not mention the sex difference in the sample (how many males/females?).
Response 10: Thank you for pointing this out. We agree with this comment. Therefore, we have changed to emphasized this point. Mention exactly where in the revised manuscript this change can be found - 12 page, 5 paragraph, and 400 line.
Comments 11: In the “methods” section, authors could indicate how long the samples remained in formalin for fixation.
Response 11: Thank you for pointing this out. We agree with this comment. Therefore, we have changed to emphasized this point. Mention exactly where in the revised manuscript this change can be found - 12 page, 6 paragraph, and 411 line.
Comments 12: Have primer efficiency tests or dissociation curve analyses been performed? Authors should explicit it in the text. The statistical analysis is described in poorly understandable terms and there is a lack of reference to threshold values ​​(e.g. p<0.05). Authors should at least explain why.
Response 12: Thank you for pointing this out. We agree with this comment. Therefore, we have changed to emphasized this point. Mention exactly where in the revised manuscript this change can be found - 14 page, 4 paragraph, and 483-486 line.
Comments 13: The detection threshold was set at 354 RFU for all genes, but it is not explained how this threshold was selected or whether it is satisfactory. The statistical description (GraphPad Prism) and how ∆Ct was calculated is unclear and fragmented, no mean values ​​± deviations, nor p-values ​​or standard errors are reported. Authors should clarify these aspects.
Response 13: Thank you for pointing this out. We agree with this comment. Therefore, we have changed to emphasized this point. Mention exactly where in the revised manuscript this change can be found - 13 page, 5 paragraph, and 465-475 line.
Comments 14: The authors state (Results 2.3) that 5S is always amplified first (Ct about 10), followed by Gapdh (~20), Sort1 and B2m (the latter Ct >30). They suggest that low Ct indicates high stability. This reasoning is sensible but not complete: a low Ct indicates more initial template, therefore 5S is more abundant. To estimate the PMI it would be more correct to use ∆Ct (normalizing to a constant reference gene) or ΔΔCt between different times. The authors calculate a ΔCt using 5S as a reference but then do not clarify how to interpret the resulting ∆Ct in terms of PMI. For example, they say that in brain the ∆Ct of Sort1 varies with time at 26°C while Gapdh and B2m, while in muscle “no PMI effect” is observed. These results are confusing: if 5S is stable, but Sort1 is decreasing, one would expect a large increase in the ΔCt of Sort1/5S. However, the discussion does not show any concrete graphs or values, making it difficult to evaluate. Authors should clarify these aspects.
Response 14: Thank you for pointing this out. We agree with this comment. Therefore, we have changed to emphasized this point. Mention exactly where in the revised manuscript this change can be found - 5 page, 4 paragraph, and 207-225 line.
Comments 15: Authors state that these biomarkers could also be used in human tissues to estimate PMI, based on relative stability, but this generalization is not completely supported. The data are obtained in 6-week-old homozygous rats, with no age variations or causes of death. Knowing that species, tissues and environmental context greatly influence degradation, it is not automatic to translate the results to the human system. For example, studies indicate that in human samples PMI does not always linearly influence RNA quality (doi: 10.1007/s00414-006-0131-9) and that muscle and brain are among the best preserved tissues. The authors should rephrase this concept.
Response 15: Thank you for pointing this out. We agree with this comment. Therefore, we have changed to emphasized this point. Mention exactly where in the revised manuscript this change can be found - 12 page, 2 paragraph, and 372-386 line.
Comments 16: There are some arbitrary statements. For example, the discussion states that 5S “prevents RNA degradation by intracellular exonucleases”. This statement is vague and biologically imprecise; it is more correct that ribosomal RNA is abundant, not that it “prevents” degradation. Furthermore, a difference in 5S Ct between 26°C and 4°C “in kidney and liver” after 48h is cited, but the study does not show kidney or liver; the authors are probably mixing their own data and literature. Such misleading details should be corrected or clarified (e.g. indicating whether they derive from previous studies or from the experimental set not shown).
Response 16: Thank you for pointing this out. We agree with this comment. Therefore, we have changed to emphasized this point. Mention exactly where in the revised manuscript this change can be found - 10 page, 1 paragraph, and 269-306 line.
Comments 17: The terms “Ct decreases” and “increases” are confusing in the text. In qPCR a lower Ct indicates more RNA, but in post-mortem data one would expect higher Ct (degradation) with increasing PMI. The paper states that “Ct values ​​of target genes decreased”, which seems to be a conceptual error. Additionally, the statistical method described for comparing Ct/∆Ct is unclear. If authors do not agree with this interpretation, they should tell me why and clarify it in the text.
Response 17: Thank you for pointing this out. We agree with this comment. Therefore, we have changed to emphasized this point. Mention exactly where in the revised manuscript this change can be found - 5 page, 2 paragraph, and 195-203 line.
Reviewer 2 Report
Comments and Suggestions for Authors
Dear Authors,
the study presents interesting and valuable findings; however, the manuscript would be improved, according to the following suggestions:
- Please clarify whether positive and negative controls were included during the RNA analysis. This is important to ensure the reliability and reproducibility of the results.
- Line 44: Consider removing the sentence “Death is defined as the loss of life.” It is redundant and not necessary in the context of a scientific manuscript.
- Line 85: Please provide a citation for the statement: “Recent forensic DNA investigations, various types of PCR-based analysis tools have been commonly used to identify DNA in very small amounts of trace evidence and RNA degradation after death is especially useful for precise estimation.”
- Line 194: Remove the extra space between “Figure 10-B” and “shows” for consistency in formatting.
- Discussion: it would benefit from a clearer outline of the study’s limitations and a more defined perspective on future research directions.
- Improve the resolution of Figures 7, 9, and 10 to enhance visual clarity.
- Review the figure numbering throughout the manuscript; Figures 2 and 8 appear to be missing or misnumbered.
- Figures 3, 4, 5, and 6 may be more appropriately included as supplementary material to streamline the main text and focus on key findings.
- To facilitate a clearer understanding of the data, consider including a comprehensive table summarizing RNA degradation results across different temperatures and time points.
Author Response
Comments 1: Please clarify whether positive and negative controls were included during the RNA analysis. This is important to ensure the reliability and reproducibility of the results.
Response 1: Thank you for pointing this out. I agree with this comment. Therefore, I have changed to emphasized this point. Mention exactly where in the revised manuscript this change can be found - 13 page, 5 paragraph, and 458-460 line.
Comments 2: Line 44: Consider removing the sentence “Death is defined as the loss of life.” It is redundant and not necessary in the context of a scientific manuscript.
Response 2: Thank you for pointing this out. I agree with this comment. Therefore, I have changed to emphasized this point. Mention exactly where in the revised manuscript this change can be found - 2 page, 1 paragraph, and 47 line.
Comments 3: Line 85: Please provide a citation for the statement: “Recent forensic DNA investigations, various types of PCR-based analysis tools have been commonly used to identify DNA in very small amounts of trace evidence and RNA degradation after death is especially useful for precise estimation.”
Response 3: Thank you for pointing this out. I agree with this comment. Therefore, I have changed to emphasized this point. Mention exactly where in the revised manuscript this change can be found - 3 page, 2 paragraph, and 107-109 line.
Comments 4: Line 194: Remove the extra space between “Figure 10-B” and “shows” for consistency in formatting.
Response 4: Thank you for pointing this out. I agree with this comment. Therefore, I have changed to emphasized this point. Mention exactly where in the revised manuscript this change can be found - 5 page, 6 paragraph, and 217 line.
Comments 5: Discussion: it would benefit from a clearer outline of the study’s limitations and a more defined perspective on future research directions.
Response 5: Thank you for pointing this out. I agree with this comment. Therefore, I have changed to emphasized this point. Mention exactly where in the revised manuscript this change can be found - 11 page, 5 paragraph, and 348-352 line.
Comments 6: Improve the resolution of Figures 7, 9, and 10 to enhance visual clarity.
Response 6: Thank you for pointing this out. I agree with this comment. Therefore, I have changed to emphasized this point. Mention exactly where in the revised manuscript this change can be found - supplementary figure S1, S2, S3.
Comments 7: Review the figure numbering throughout the manuscript; Figures 2 and 8 appear to be missing or misnumbered.
Response 7: Thank you for pointing this out. I agree with this comment. Therefore, I have changed to emphasized this point. Mention exactly where in the revised manuscript this change can be found - 7 page, 1 paragraph, and 234 line.
Comments 8: Figures 3, 4, 5, and 6 may be more appropriately included as supplementary material to streamline the main text and focus on key findings.
Response 8: Thank you for pointing this out. I agree with this comment. Therefore, I have changed to emphasized this point. Mention exactly where in the revised manuscript this change can be found - Supplementary figure S1, S2, S3.
Comments 9: To facilitate a clearer understanding of the data, consider including a comprehensive table summarizing RNA degradation results across different temperatures and time points.
Response 9: Thank you for pointing this out. I agree with this comment. Therefore, I have changed to emphasized this point. Mention exactly where in the revised manuscript this change can be found - Supplementary table S1, S2, S3, S4, S5, S6, S7.
Reviewer 3 Report
Comments and Suggestions for Authors
The article is consistent with the Journal aim and addresses a relevant topic for forensic sciences, PMI definition, through experimental rat models. The methods are clear as the overall text.
References are adequate and consistent.
I have some minor points to be kindly addressed:
.introduction, line 52 please discuss briefly and reference main methods for PMI estimation highlighting those molecular/genomic ones. Highlight and reference difference in tissue degradation and then adequacy/fitness to genomic investigations between muscle and brain.
. discussion line 216-236 seems methods, please revise.
. add a conclusive paragraph to strengthen implications and future research also on human bodies
Author Response
Comments 1: introduction, line 52 please discuss briefly and reference main methods for PMI estimation highlighting those molecular/genomic ones. Highlight and reference difference in tissue degradation and then adequacy/fitness to genomic investigations between muscle and brain.
Response 1: Thank you for pointing this out. We agree with this comment. Therefore, we have changed to emphasized this point. Mention exactly where in the revised manuscript this change can be found - 2 page, 3 paragraph, and 73-80 line.
Comments 2: discussion line 216-236 seems methods, please revise.
Response 2: Thank you for pointing this out. We agree with this comment. Therefore, we have changed to emphasized this point. Mention exactly where in the revised manuscript this change can be found - 10 page, 1 paragraph, and 269-291 line.
Comments 3: add a conclusive paragraph to strengthen implications and future research also on human bodies
Response 3: Thank you for pointing this out. We agree with this comment. Therefore, we have changed to emphasized this point. Mention exactly where in the revised manuscript this change can be found - 12 page, 4 paragraph, and 387-395 line.
Round 2
Reviewer 1 Report
Comments and Suggestions for Authors
Dear authors,
Thank you for your efforts to improve the manuscript based on the comments provided. A few final improvements are needed.
Some references are very dated and do not always reflect what is reported in the manuscript (e.g., references 5, 25 and 30). Authors should update the references.
Comments on the Quality of English LanguageEnglish still needs a thorough revision. There are still several errors, which reduce the text readability.
Author Response
Comments 1: Some references are very dated and do not always reflect what is reported in the manuscript (e.g., references 5, 25 and 30). Authors should update the references.
Response 1: Thank you for pointing this out. We agree with this comment. Therefore, we have changed to emphasized this point. Mention exactly where in the revised manuscript this change can be found - 15 page, 2 paragraph, and 515 line. 16 page, 1 paragraph, and 550-551 line. 16 page, 1 paragraph, and 560-561 line.